# Microstructure Evolution of 316L Steel Prepared with the Use of Additive and Conventional Methods and Subjected to Dynamic Loads: A Comparative Study

**DOI:** 10.3390/ma13214893

**Published:** 2020-10-31

**Authors:** Michał Ziętala, Tomasz Durejko, Robert Panowicz, Marcin Konarzewski

**Affiliations:** 1Department of Materials Technology, Military University of Technology, Gen. Kaliskiego Str. 2, 00-908 Warsaw, Poland; michal.zietala@wat.edu.pl (M.Z.); tomasz.durejko@wat.edu.pl (T.D.); 2Faculty of Mechanical Engineering, Military University of Technology, Gen. Kaliskiego Str. 2, 00-908 Warsaw, Poland; marcin.konarzewski@wat.edu.pl

**Keywords:** 316L stainless steel, split Hopkinson pressure bar, high-strain-rate testing, additive manufacturing, laser engineered net shaping, structure, EBSD

## Abstract

The mechanical properties and microstructure evolution caused by dynamic loads of 316L stainless steel, fabricated using the Laser Engineered Net Shaping (LENS) technique and hot forging method were studied. Full-density samples, without cracks made of 316L stainless steel alloy powder by using the LENS technique, are characterized by an untypical bi-modal microstructure consisting of macro-grains, which form sub-grains with a similar crystallographic orientation. Wrought stainless steel 316L has an initial equiaxed and one-phase structure, which is formed by austenite grains. The electron backscattered diffraction (EBSD) technique was used to illustrate changes in the microstructure of SS316L after it was subjected to dynamic loads, and it was revealed that for both samples, the grain refinement increases as the deformation rate increases. However, in the case of SS316L samples made by LENS, the share of low-angle boundaries (sub-grains) decreases, and the share of high-angle boundaries (grains of austenite) increases. Dynamically deformed wrought SS316L is characterized by the reverse trend: a decrease in the share of high-angle boundaries and an increase in the share of low-angle boundaries. Moreover, additively manufactured SS316L is characterized by lower plastic flow stresses compared with hot-forged steel, which is caused by the finer microstructure of wrought samples relative to that of additive samples. In the case of additively manufactured 316L steel samples subjected to a dynamic load, plastic deformation occurs predominantly through dislocation slip, in contrast to the wrought samples, in which the dominant mechanism of deformation is twinning, which is favored by a high deformation speed and low stacking fault energy (SFE) for austenite.

## 1. Introduction

Nowadays, there is rapid development of additive techniques for manufacturing structural elements. The use of these techniques allows a prototype of the designed element to be obtained quickly and relatively cheaply, without the need to use virtually any mechanical treatment (cutting, milling, drilling, etc.). Currently, Fused Deposition Modeling (FDM) technology is used on a massive scale, which enables the production of finished elements from various types of plastics [1]. It is also possible to use metal alloys for additive manufacturing (AM); however, because of the mechanical properties of this material group (particularly a high melting point), it was necessary to develop technologies other than those used for plastics. Using the method of feeding or joining the material, a number of different techniques can be distinguished, the most popular of which are Laser Metal Deposition (LMD), Selective Laser Sintering (SLS), Selective Laser Melting (SLM), and Laser Engineered Net Shaping (LENS) [2,3,4]. The LENS technology poses a few essential advantages over other methods, such as ability to fabricate components, which are almost ready to use and can be manufactured using a wide variety of metals and alloys. Moreover, easier control of the structure is possible, compared to other methods [5,6].The results obtained by OPTOMEC company (manufacturer of LENS, Albuquerque, NM, USA), indicate that additive manufactured 316 stainless steel manufactured with use of their device is characterized by a fine-grain cellular microstructure (grains size of several microns), with yield strength twice that of conventionally shaped SS316L [7]. 

Moreover, the precise control of technological parameters of LENS process, i.e., the amount of powder, laser power, feed of a working table, and focusing point position, became a powerful tool for the microstructure development of manufactured components.

Additionally, Keicher [8] produced stainless steel samples using LENS technique, which were characterized by ultimate tensile strength (UTS) of 590 MPa, a yield strength (YS) of 240 MPa, and a total elongation of 50%. In [9], the authors showed that by using the LENS process, it is possible to produce a full-density cubic sample without structural defects from 316L stainless steel powder prepared through gas atomization. Microscopic observations of the structure (SEM) and an electron backscattered diffraction (EBSD) analysis revealed, that additively manufactured samples are characterized by fine-grain microstructure consisting of elongated austenite grains with size ~5 µm. Moreover, transmission electron microscopy analysis showed that the dislocations are present only inside the austenite cells and not in the intercellular ferrite boundaries. Additive manufactured SS316L by LENS is characterized by better mechanical properties than steel fabricated by conventional methods. The YS and UTS measured both perpendicular and parallel to the building direction are comparable to the results obtained for other additive manufacturing methods with laser source for powder melting and significantly higher than for SS316L shaped by casting and plastic forming.

Many construction materials, including 316L steel, are highly sensitive to the strain rate. There are only a few methods that can be used to test material properties at a high strain rate, and the most common in the literature is the method that employs a modified Hopkinson bar [10,11,12,13]. Publications on the dynamic properties of materials mainly focus on the macroscopic description of the effect of the strain rate on material behavior. Therefore, there are relatively few articles that describe the impact of the strain rate on the mechanical properties and microstructure of materials, particularly those produced using additive techniques. Lee et al. [12] conducted tests on samples prepared from sintered 316L powder at strain rates ranging from 2.7 × 10^3^ s^−1^ to 7.5 × 10^3^ s^−1^. During their research, they observed that a change in the strain rate also changed the microstructure of the sample: samples damaged at a higher strain rate were characterized by lower porosity and grain size. These changes are related to the fact that as the strain rate increases, the probability of dynamic recrystallization also increases [14,15]. The results for the tested samples prepared using the SLM method can be found in [13]. The authors conducted tests on both solid cylindrical and cubic lattice samples made from 316L steel, produced using the additive method. The research was carried out in both a static and a dynamic range using the split Hopkinson pressure bar (SHPB) method with an average strain rate of 5 × 10^2^ s^−1^. A microstructural analysis of the tested samples was also carried out. In the microscopic image, the porosities of strut surfaces and lattice nodes, as well as irregularities in the shape of grains elongated along the strut axes, were observed. However, this analysis was only carried out for cubic lattice specimens that were not subjected to experimental tests.

The purpose of this article is to present a comparative study of the structure evolution of 316L steel prepared with the use of additive and wrought methods and subjected to a dynamic load. The SHPB method was used for dynamic tests, and the structure evolution was analyzed using a field emission gun scanning electron microscope (FEG SEM), FEI Quanta 3D (Hillsboro, OR, USA). The microscope was equipped with an automatic system for analyzing the electron backscatter diffraction (EBSD) produced by the TSL Company (Draper, UT, USA). The EBSD technique, with a spatial resolution of 30–50 nm [16], was used for both qualitative and quantitative identification of the microstructure (grain boundary disorientation, grain orientation, as well as grain size and shape).

The SS316L wrought alloy is used in a growing number of industries (automotive, aviation, nuclear, chemical) due to its very good mechanical parameters as well as high resistance to environmental factors, mainly high resistance to corrosion [17]. Due to the abovementioned factors and good biocompatibility, this material is also used for implants [18]. Furthermore, elements prepared from this steel are frequently subjected to dynamic loads during their service lives. The increasing use of additive methods for manufacturing not only visual prototypes, but also finished products makes it necessary to know the material properties of such elements in both static and dynamic load range. In the literature, we can find only three articles [19,20,21] concerning the behavior of the SS316L wrought alloy under dynamic load, while there is no information about additively manufactured 316L tests in this load range. We hope that the presented article will at least partially fill the described gap in research results.

## 2. Materials and Methods

### 2.1. Material and Specimen Preparation

The material used in this research was 316L stainless steel. The samples were prepared by both additive and conventional methods.

In the case of the additive method, the LENS process and material in powder form were utilized. The chemical composition of the powder that is commonly used in industry is listed in Table 1. Moreover, in this table, the chemical composition of LENS sample and commercial implant was also shown. Obtained results for both powder and samples subjected to dynamic loads were similar to each other and are corresponding to the nominal values.

Used SS316L gas atomized powder was produced by LPW Technology (Runcorn, England). Its shape was spherical shape with particle size in the range of 44–150 μm. A cylindrical sample with a diameter of 30 mm and a height of 40 mm was produced from SS316L steel plate (200 mm × 200 mm × 6.3 mm), with the processing parameters shown in Table 2.

In the case of the conventional method, a commercial hot forged hip implant FENIX 5 manufactured by PPUH Medgal company (Białystok, Poland) was used.

All of the samples for the SHPB experiments, with dimensions of L = 2, 4 and 6 mm and a diameter of 4 mm, were cut using a wire electro-discharge machining (WEDM) device, which ensured a small diameter distribution of the samples of 0.01 mm.

Metallographic preparation was conducted for all samples using the following order: grinding on SiC papers with granulations of 120–2400, polishing with diamond suspensions (3 µm, 1 µm, and 0.25 µm), and polishing with silica suspensions of 0.1 µm and 0.06 µm (to prepare samples for EBSD analysis).

### 2.2. SHPB Experiments

Tests in high strain rate conditions were performed at room temperature with the use of a classical split Hopkinson compression bar technique (Figure 1). The testing system was composed of bars made from maraging high-strength steel (Young’s modulus E = 184 GPa, speed of sound C = 4755 m/s). The incident and transmission bars had a diameter of 10 mm and length of 1000 mm, and the striker bar had a length of 100 mm and the same diameter as the other bars. The striker bar was driven by a compressed air system to speeds ranging from 10 to 18 m/s.

In order to limit Pochhammer–Chree high-frequency oscillations, the front of the striker bar, which hits the input bar, was rounded [22,23,24,25], whereas the end faces of the specimens were lubricated with a silicon paste before the experiments to ensure a uniaxial deformation state (frictionless contact between bars and specimen).

The signals propagating in the bars were recorded with the use of strain gauges working in a typical half-bridge configuration, located on the opposite sides of the bars. Locating the strain gauges on both sides of the bar minimizes disturbances in the measured signals that originate from the propagation of bending waves in the bars [26]. Electrical strain gauges CEA-13-062UW-350 (gauge length 1.57 mm) made by Vishay Micro-Measurements were used in the SHPB tests. The signals were recorded by the 24-bit ADC multi-functional data acquisition device LTT24 from LTT Labortechnik Tasler GmbH (Würzburg, Germany), which sampled the signal at 4 MHz. It enabled the measurements of small-amplitude signals without the use of amplifiers that limit the bandwidth.

The plastic flow stress, strain and strain rate of the specimen material were determined according to the classical three-way theory (1)–(3) based on the recording signals [23]:(1)$$\sigma \left(t\right)=\frac{A}{2A_{S}}E\left[\varepsilon_{I}\left(t\right)+\varepsilon_{R}\left(t\right)+\varepsilon_{T}\left(t\right)\right]$$
(2)$$\dot{\varepsilon }\left(t\right)=\frac{C}{L_{S}}\left[\varepsilon_{I}\left(t\right)-\varepsilon_{R}\left(t\right)-\varepsilon_{T}\left(t\right)\right]$$
(3)$$\varepsilon \left(t\right)=\frac{C}{L_{S}}\overset{t}{\underset{0}{\int }}\left[\varepsilon_{I}\left(\tau \right)-\varepsilon_{R}\left(\tau \right)-\varepsilon_{T}\left(\tau \right)\right]d\tau$$
where *A_s_* is the initial cross-sectional area, *C* is sound velocity, *L_s_* is length of the specimen, *E* is Young’s modulus, ε(*t*) is the strain signal from the strain gauges, *A* is the initial cross-section of the bars, and ε(*t*) is the strain signal from the strain gauges. The subscripts *I*, *R* and *T* denote the incident, reflected, and transmitted pulses.

The true stress, strain and strain rate data were obtained using Equations (4)–(6), respectively [16]:(4)$$\sigma_{T}=\sigma \left[1-\varepsilon \right],$$
(5)$${\dot{\varepsilon }}_{T}=\frac{\dot{\varepsilon }}{1-\varepsilon },$$
(6)$$\varepsilon_{T}=-\ln\left[1-\varepsilon \right].$$

During dynamical deformation, the temperature rise caused by the conversion of plastic work into heat is described by Equation (7) [27]:(7)$$\Delta T=\frac{\eta }{\varrho c_{p}}\overset{\varepsilon_{p}}{\underset{0}{\int }}\sigma \left(\varepsilon \right)d\varepsilon$$
where p is density, *c_p_* is heat capacity (*c_p_* = 500 J/kg K [28]), σ is stress, *d*ε is strain interval, and η = 0.9 is the Taylor–Quinney parameter, which describes the conversion rate of mechanical energy into heat [8,10].

The temperature rise caused by plastic deformation depends on all parameters/variables that affect the value of the plastic stress flow of the material. Therefore, the temperature increases with an increase in the deformation rate and decreases with an increase in the temperature.

### 2.3. Microstructural Investigation

The electron backscattered diffraction technique was used for qualitative and quantitative microstructure measurements. Parameters, such as grain sizes and shapes, grain boundary misorientation, and grain orientation were analyzed. Additionally, a strain/stress analysis based on Kernel Average Misorientation (KAM) parameter distribution maps was performed. 

The KAM parameter is an averaged measure of disorientation of a given grain in relation to neighboring grains and it is used for the determination of the misorientation θ angle value. It has to be calculated with a fixed neighbor distance, not including all points of the kernel. KAM parameter is a good indicator of the local misorientation, thereby a good indicator of Geometrically Necessary Dislocations (GND) density. GND [29] refers to the dislocations which are able to accommodate the lattice curvature taking place during the plastic deformation of materials as a result of non-uniform strain at the crystal scale [30]. EBSD data were obtained using an hkl diffraction index and all disorientation angles between grains > 15° are considered to be high-angle boundaries.

The tests were carried out on samples whose orientation was related to Kikuchi’s diffraction condition. The samples were placed in a special holder to maintain a 70° angle between the surface of the sample and the CCD camera (Hillsboro, OR, USA). The fully automatic EBSD system, using a sensitive camera, recorded diffraction images in the form of diffraction bands (the so-called Kikuchi lines that result from the deflection of the electron beam on crystallographic planes) that are characteristic of individual crystallographic structures. This system allows for the analysis of relatively large areas of the sample in a short time, recording and indicating about 50,000 Kikuchi line systems within 10 min.

## 3. Results

### 3.1. Mechanical Properties

First, the tests were carried out on 316L steel samples made using LENS technology under the same test conditions. From the results of the study, stress and strain values were estimated according to Equations (1)–(6), which allowed for the development of true stress–strain curves, as presented in Figure 2, at an average strain rate of 2840 s^−1^. It was found that the dispersion in stress values increases slightly with an increase in deformations and is comparable to the material obtained with the classical method. Plastic flow stress values for selected deformations of 5% and 15% are presented in Table 3.

In the next step, the wrought samples of SS316L and SS316L manufactured with LENS technology were tested at different strain rates (Figure 3).

The presented results show that the value of plastic flow stress of the SS316L wrought alloy is more than 300 MPa higher than in the case of additively manufactured 316L for strain rates lower than about 1000 s^−1^. This difference increases with increasing strain rates, reaching almost 600 MPa at a strain rate of about 3000 s^−1^. This means that the SS316L wrought alloy is more sensitive to dynamic loads than the material made using the additive method. The difference in the strain rate sensitivity of these two materials is about 2, however, the relationship between the plastic flow stress of the SS316L wrought alloy and AM 316L is an almost linear function of the strain rate for ε = 3% (Table 4).

The increase in the temperature of the samples after the tests was estimated from Equation (7) and ranged from 9 to 62 °C. However, the duration time of the load is so short that the material does not recover. Additionally, analyzing the iron-carbon phase diagram we can observe that phase transition can occur only above the temperature of 230° C. Therefore, the evolution process of the structure of the considered materials under dynamic loads is not affected by the temperature.

### 3.2. Microstructure Evaluation

As a result of dynamic plastic deformation in 316L steel manufactured with the laser additive technique, grain growth is observed at low deformation rates of about 1000 s^−1^ (Figure 4b,c and Figure 5a) In contrast, a further increase in the deformation rate results in grain refinement (Figure 4d–f and Figure 5a). This results from the fact that at low deformation rates, dynamic recovery takes place, and the original structure (layered and with grains of a maximum thickness of about ~0.3 mm) of additively manufactured 316L disappears (Figure 4a and Figure 6a). It should also be noted that immediately after the LENS process, the 316L steel is in a state that is characterized by a bi-modal structure consisting of macro-grains (microstructure), which include sub-grains with a similar crystallographic orientation (dislocation substructure). This is also confirmed by the share of low- and high-angle boundaries: in the 316L steel, immediately after the LENS process, the share of high-angle boundaries is more than twofold greater than that of low-angle boundaries (Figure 6a and Figure 7a).

Plastic deformation at low speeds (970 s^−1^ and 1180 s^−1^) causes the shares of low- and high-angle boundaries in the material to invert, which is due to a significant increase in dislocation as a result of dynamic deformation and polygonization. The obtained results also indicate that the plastic deformation takes place along with the participation of local rotations of the crystal lattice in the volume and on the boundaries of the individual macro-grains. In 316L steel, plastic deformation is caused by a slipping or twinning process, which leads to the rotation of the crystal lattice of adjacent micro-areas. Low-angle boundaries that separate dislocation cells and sub-grains, as well as macro-grains consisting of sub-grains of similar crystallographic orientation (as a result of the rotation of the crystal lattice), form at low deformations. Conversely, in a highly plastically deformed material (for high strain rates), the adjacent micro-areas (sub-grains) exhibit a relatively large disorientation angle to each other, which leads to a significant increase in the share of high-angle boundaries (Figure 4d–f and Figure 7a) and significant grain fragmentation (Figure 5a).

The above theses are confirmed by the imaging of the KAM parameter (Figure 8), which defines the disorientation of individual micro-areas. It is observed that with an increase in a dynamic deformation rate, the share of grains in which the KAM parameter value reaches 3–5° increases significantly. In addition, there is a visible change in the microstructure morphology of additively manufactured 316L (Figure 6a) subjected to dynamic plastic deformation (Figure 6b–f). As a result, at low rates of deformation, the primary grains grow to form macro-grains that range from 400 µm to 850 µm. Increasing the deformation rate, on the other hand, leads to grain fragmentation (Figure 6d–f); grains with sizes of up to 150 µm constitute over 60% of the total grain population.

In the case of 316L hot-forged steel subjected to dynamic plastic deformation, the grain size of the reference sample, as well as that of the samples subjected to plastic deformation, is much smaller than it is in the case of the additively manufactured 316L steel (Figure 5b). In the case of the reference sample, 5% of the surface of the analyzed specimens corresponds to a grain size in the range of 45–65 µm.

For the reference sample and samples deformed at speeds of $\dot{\varepsilon }$ = 620 s^−1^, $\dot{\varepsilon }$ = 830 s^−1^ and $\dot{\varepsilon }$ = 970 s^−1^, the share of grains with a size in the range of 0–15 µm is equal to 41%, 47%, 44%, and 48%, respectively. A slight increase in grain fragmentation is observed with an increase in the deformation rate. A noticeable increase in grain refinement occurs when the forged sample of 316L steel is deformed at a rate of $\dot{\varepsilon }$ = 3260 s^−1^. In this case, the share of 0–15 µm grains increases to 74%. This may indicate that when 316L steel is shaped using the conventional method and then subjected to dynamic plastic deformation with a deformation rate of $\dot{\varepsilon }$ = 3260 s^−1^, its structure undergoes fragmentation (Figure 9e) as a result of high plastic deformation.

In the case of samples produced with the use of the hot forging method, an increase in the share of low-angle boundaries is also observed along with an increase in the strain rate (Figure 7b), in contrast, to additively manufactured 316L steel (Figure 7a). This indicates that during dynamic plastic deformation of the hot-forged steel, in addition to dynamic recovery, which leads to fragmentation of the structure, there is also a process of polygonization that leads to a significant increase in the share of low-angle boundaries (Figure 10) and thus in the formation of sub-grains, whose walls are composed of low-angle boundaries with a high dislocation density. The deformation rate increases with an increase in the value of the KAM parameter, as in the case of additively manufactured 316L steel. However, in 316L hot-forged steel, i.e., in the initial state, the amount of stress (understood as the value of the KAM parameter > 1°) is minimal (Figure 10) in contrast to AM 316L steel, for which the material in the initial state (directly after the LENS process) is characterized by a significant share of internal stresses (a significant share of areas where the KAM parameter is in the range of 1–5°; Figure 8a).

## 4. Discussion

When 316L steel is produced by the LENS technique, it undergoes plastic flow at lower stress values than hot-forged steel (Table 4). This is observed because hot-forged steel is characterized by a finer microstructure in the initial state (average grain size D_avg_ ~ 20 µm) compared with additively manufactured 316L (D_avg_ ~ 137 µm). According to the Hall–Petch relationship, the higher the flow stress, the finer the grain. The fine-grained structure seen in additively manufactured 316L steel is a substructure formed by low-angle (dislocation) boundaries, and fine sub-grains are grouped into macro-grains defined by high-angle boundaries. Those micro- and macrostructures (high dislocations density) have significant effect on the plastic deformation mechanism, which is a slip of dislocations. It is in contrast to the hot forged 316L stainless steel subjected to dynamic loads, in which way of shaping leads to forming an equiaxed and stress-free microstructure. A similar state of the material was described in the work [19], where authors presented a 316L stainless steel rod subjected to annealing and saturation before plastic deformation.

The second factor that conditions the plastic flow of additively manufactured SS316L is that the steel undergoes a cyclical heating and cooling process during manufacturing, which leads to both recovery and an increase in the ferrite content at grain boundaries, i.e., a phase that is characterized by nearly 2 times less tensile strength than that of austenite. Figure 11 shows the microstructure of the 316L steel samples—classically shaped (Figure 11a,b) and produced by the LENS technique (Figure 11c,d)—after they were subjected to dynamic plastic deformation at the minimum and maximum strain rates. It is clearly seen that in the case of additively manufactured 316L steel samples subjected to dynamic deformation, this deformation occurs predominantly through dislocation slip, as shown by the clearly visible slip/shear bands on the surface of the tested samples, both for the minimum and maximum strain rates (Figure 11c,d).

In the case of the hot-forged 316L steel subjected to dynamic plastic deformation, with the maximum ε = 0.2, the dominant deformation mechanism is twinning, which is favored by a high deformation speed and low stacking fault energy (SFE) for austenite. In the work [31], authors showed a four-stage strain-hardening response model for low SFE materials (like 316L stainless steel). Based on the research, they found that they were only above the degree ε = 0.2 number of deformation twins for 316L steel decreases, and distinct shear bands appear in the volume of austenite grains. On the other hand, Eskandari and Szpunar [32] subjected the high manganese austenitic steel to dynamic impact loading, and they showed that for the strain rates above 1 × 10^3^ s^−1^ or 2 × 10^3^ s^−1^ [33], the twinning does not occur. They explain this by a significant increase in stacking fault energy with an increase of strain rate (twinning is more possible for lower SFE).

Based on Byun’s equation, the authors of the work [34] indicated, that the value of the critical stresses necessary to initiate the deformation by twinning increase with increasing temperature For 25 °C and 100 °C critical stresses are equal to ~400 MPa and ~540 MPa, respectively. In our case, the average value of stresses is ~810MPa, and more, then, is needed to initiate the twinning mechanism. Moreover, basing on the EBSD analysis, they showed that all grains were oriented in such a way that was preferable for the twinning deformation mechanism.

In addition, the deformation of the forming twins leads to significant fragmentation of the structure and then to a strong strengthening of the material, which, in turn, leads to higher values of the plastic flow stress of hot-forged 316L steel. Moreover, the increase of the plastic flow stress is connected with the increase of the strain rate, which is also caused by an increase of dislocations density. A similar effect was described in the work [19] where authors subjected the 316L stainless steel rod to dynamic loads (with use of the SHPB), which was previously annealed and quenched in water (in order to remove the residual stresses). Comparing the results for the constant temperature, they observed a rapid increase of flow stress at strain rates >1 × 10^3^ s^−1^ caused by the presence of various deformation mechanisms such as a change in the rate controlling mechanism, the generation of dislocations, and the nucleation of mechanical twins and martensite transformation. In our case, we observed similar mechanisms, except the martensite transformation-before and after dynamic loads the structure consists of one austenite phase. It can be explained by that the maximum strain rate is $\dot{\varepsilon }$ = 3260 s^−1^, and the martensite transformation is induced by much higher strain rates. The increase of the plastic flow rate is also a macroscopic symptom of work-hardening with the increase of strain rate and it was shown by Lee et al. [20] and it is also explained by the twinning and the increase number of shear bands within individual grains. The shear bands limit the movement of dislocation, causing dislocation loops to form around them. Consequently, the material strengthens more intensively as the strain rate increases at a lower temperature.

Lee et al. in [19] investigated the 316L stainless steel rod subjected to dynamic load at 25 °C and 800 °C temperature with 1 × 10^3^ s^−1^ and 5 × 10^3^ s^−1^ strain rate. Only at 5 × 10^3^ s^−1^ strain rate and high temperature author observed the regions of microstructure, where dynamic recrystallization took place. Therefore, in our research, we did not observed dynamic recrystallization phenomena, because of the implemented experimental conditions (medium strain rate and room temperature). Both continuous dynamic recrystallization (cDRX) and discontinuous dynamic recrystallization (dDRX) occur at ε > 1 and at ε > 3, respectively [35].

## 5. Conclusions

This work focuses on dynamic mechanical properties and microstructural changes during dynamic deformation of 316L produced by hot forging and additive (LENS) methods. The major conclusions are as follows:The plastic flow stress (σ**_3%_**) is significantly lower for additively manufactured 316L than that obtained by hot forging. This is a strictly microstructural effect, i.e., SS316L produced by the LENS technique is cyclically heated and cooled during production, which leads to both recovery and an increase in the ferrite content at grain boundaries, i.e., a phase that is characterized by nearly two times less tensile strength than that of austenite when the 316L fabricated with hot forging is characterized by a “pure” austenite structure.The morphology of the LENS 316L microstructure significantly changes with increasing deformation rates, i.e., the primary grains grow to form macro-grains and undergo fragmentation at low and high deformation rates, respectively. The influence of the deformation rate, by contrast, has a negligible effect on hot-forged 316L microstructure morphology.The participation of grain boundaries below 15° disorientation for hot-forged 316L increases with the increasing strain rate, opposite to the relationship in LENS 316L.Different mechanisms of deformation for hot-forged 316L and LENS 316L samples were observed. In the first case, the dominant deformation mechanism is twinning, which is favored by a high deformation speed and low SFE positioning error for austenite. On the contrary, deformation occurs predominantly through dislocation slip (clearly visible slip/shear bands on the surface of the tested samples, both for the minimum and maximum strain rates) for LENS 316L samples.

## Figures and Tables

**Figure 1 materials-13-04893-f001:**
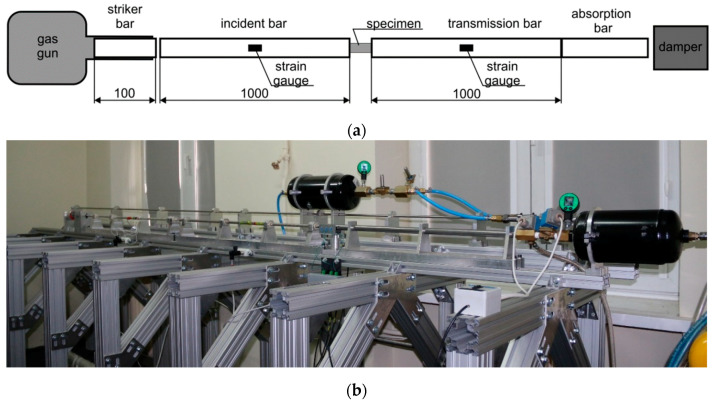
Split Hopkinson pressure bar arrangement; (**a**) schematic illustration; (**b**) experimental setup.

**Figure 2 materials-13-04893-f002:**
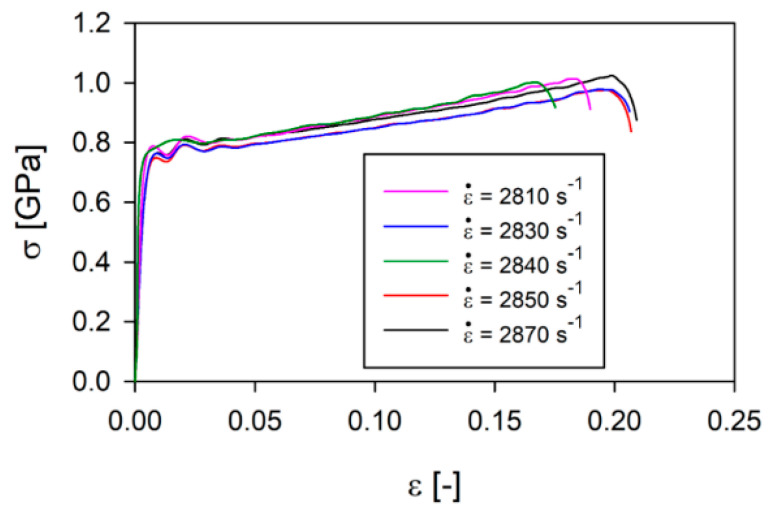
True stress–strain curves of 316L alloy made by LENS technology at an average strain rate of 2840 s^−1^.

**Figure 3 materials-13-04893-f003:**
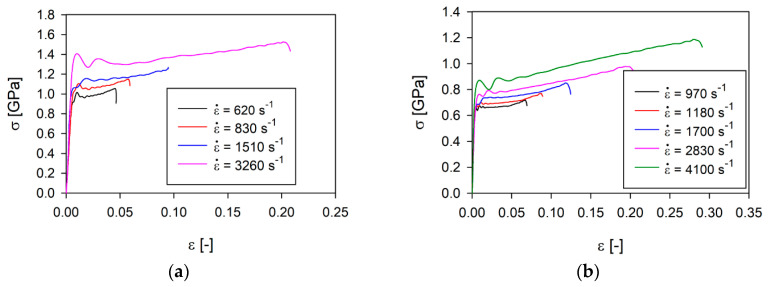
True stress–strain curves of SS316L alloy: (**a**) wrought; (**b**) prepared by LENS technology.

**Figure 4 materials-13-04893-f004:**
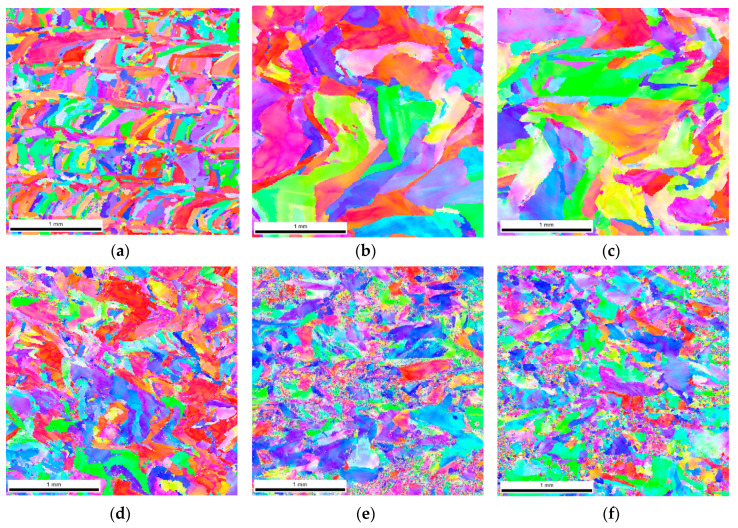
The microstructure illustrated with the reverse map of polar figures obtained with the electron backscattered diffraction (EBSD) method for additively manufactured 316L subjected to dynamic plastic deformation at various rates: (**a**) not deformed; (**b**) ε = 970 s^−1^; (**c**) ε = 1180 s^−1^; (**d**) ε = 1700 s^−1^; (**e**) ε = 2830 s^−1^; (**f**) ε = 4100 s^−1^.

**Figure 5 materials-13-04893-f005:**
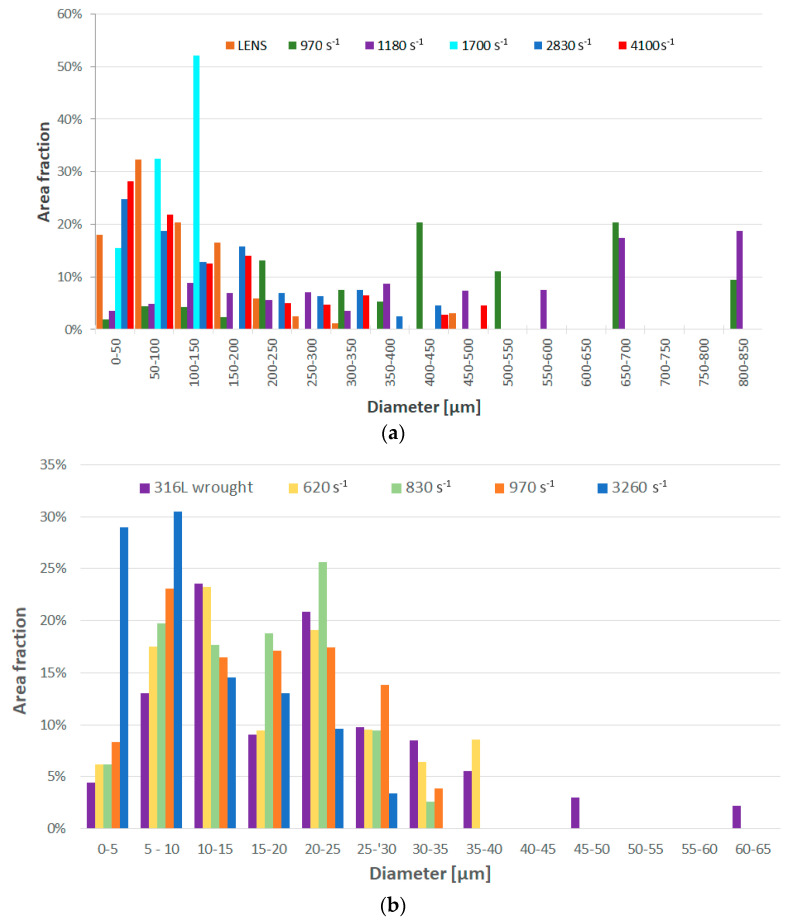
Grain size distribution: (**a**) 316L samples directly after the LENS process; (**b**) SS316L alloy subjected to dynamic deformation.

**Figure 6 materials-13-04893-f006:**
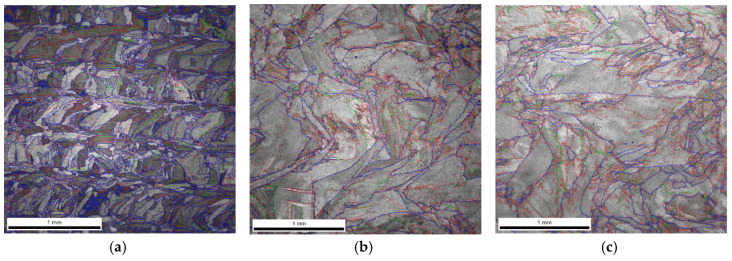
Image Quality (IQ) EBSD maps obtained for additively manufactured 316L steel samples subjected to dynamic plastic deformation at various rates: (**a**) not deformed; (**b**) $\dot{\varepsilon }$ = 970 s^−1^; (**c**) $\dot{\varepsilon }$ = 1180 s^−1^; (**d**) $\dot{\varepsilon }$ = 1700 s^−1^; (**e**) $\dot{\varepsilon }$ = 2830 s^−1^; (**f**) $\dot{\varepsilon }$ = 4100 s^−1^.

**Figure 7 materials-13-04893-f007:**
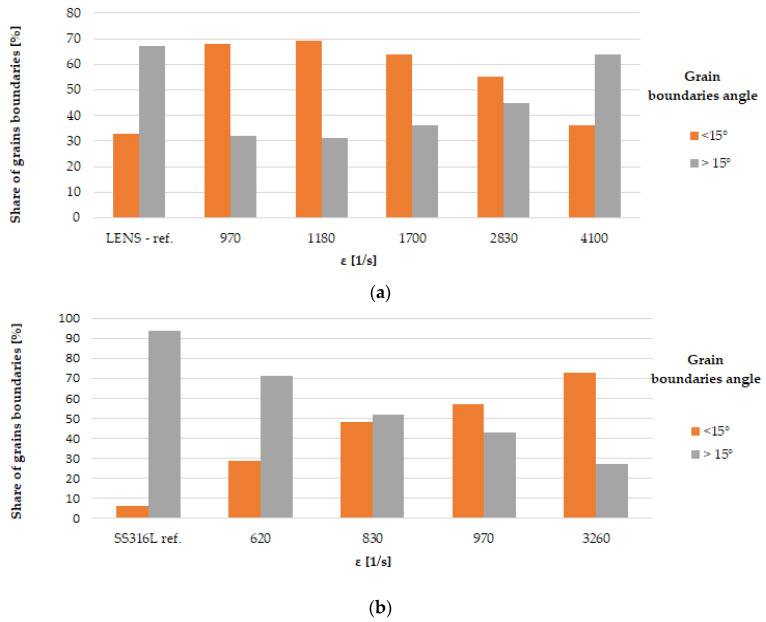
Distribution of the share of low and high angles for the 316L steel samples subjected to dynamic plastic deformation: (**a**) manufactured with the LENS method; (**b**) manufactured with the hot forging method.

**Figure 8 materials-13-04893-f008:**
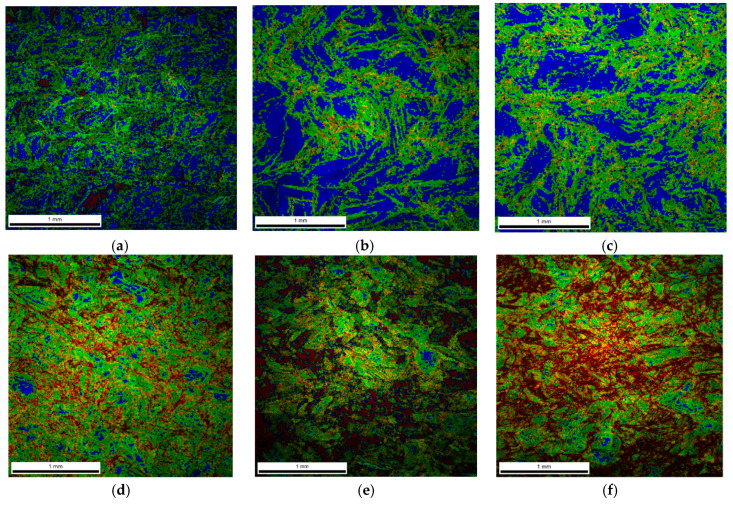
Kernel Average Misorientation (KAM) EBSD maps of samples made of additively manufactured 316L steel subjected to dynamic plastic deformation at various rates: (**a**) not deformed; (**b**) $\dot{\varepsilon }$ = 970 s^−1^; (**c**) $\dot{\varepsilon }$ = 1180 s^−1^; (**d**) $\dot{\varepsilon }$ = 1700 s^−1^; (**e**) $\dot{\varepsilon }$ = 2830 s^−1^; (**f**) $\dot{\varepsilon }$ = 4100 s^−1^.

**Figure 9 materials-13-04893-f009:**
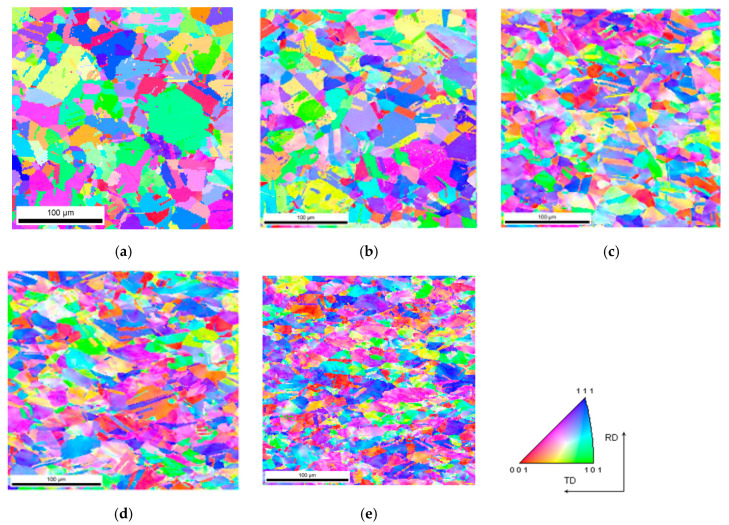
The microstructure illustrated by reverse maps of polar figures obtained with the EBSD method for an implant made of 316L steel with hot forging and subjected to dynamic plastic deformation at various rates: (**a**) not deformed: (**b**) $\dot{\varepsilon }$ = 620 s^−1^; (**c**) $\dot{\varepsilon }$ = 830 s^−1^; (**d**) $\dot{\varepsilon }$ = 970 s^−1^; (**e**) $\dot{\varepsilon }$ = 3260 s^−1^.

**Figure 10 materials-13-04893-f010:**
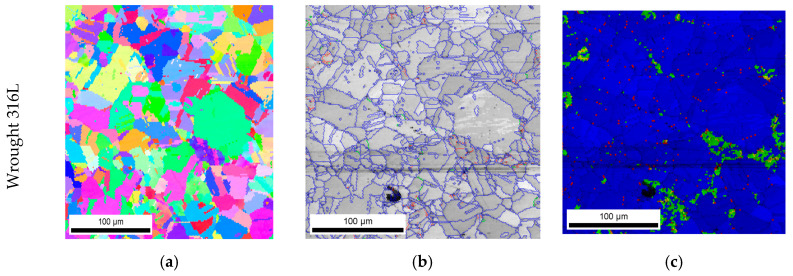
Comparison of the impact of dynamic plastic deformation on the microstructure of an implant made of 316L and subjected to dynamic plastic deformation at various rates: (**a**–**c**) not deformed; (**d**–**f**) $\dot{\varepsilon }$ = 620 s^−1^; (**g**–**i**) $\dot{\varepsilon }$ = 970 s^−1^; (**j**–**l**) $\dot{\varepsilon }$ = 3260 s^−1^. The microstructure illustrated with the reverse maps of polar figures obtained by the EBSD method (**a**,**d**,**g**,**j**). IQ EBSD maps (**b**,**e**,**h**,**k**). KAM parameter maps obtained by the EBSD method (**c**,**f**,**i**,**l**).

**Figure 11 materials-13-04893-f011:**
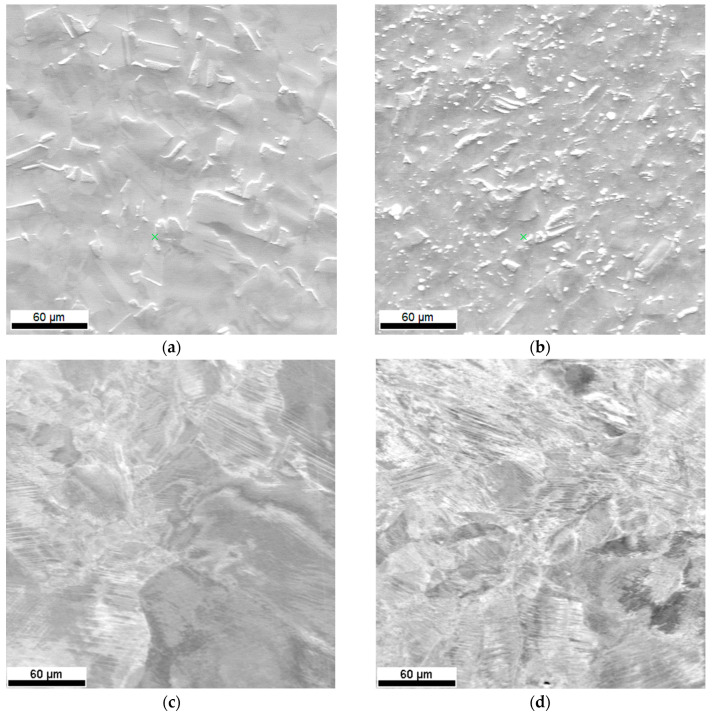
Comparison of the impact of dynamic plastic deformation on the microstructure of 316L hot-forged steel (**a**,**b**) and LENS (**c**,**d**), subjected to plastic deformation at a rate of (**a**) ε = 620 s^−1^, (**b**) ε = 3260 s^−1^, (**c**) ε = 970 s^−1^, (**d**) ε = 4100 s^−1^.

**Table 1 materials-13-04893-t001:** Chemical composition of 316L: stainless steel powder, additive manufacturing sample, and commercial implant.

Chemical Composition (wt. %)						
	C	Cr	Ni	Mn	Mo	Fe
Nominal	≤0.03	17.5	11.5	≤2	2–3	balance
Powder	*	16.84 ± 0.13	12.83 ± 0.1	1.91 ± 0.02	1.99 ± 0.02	balance
LENS sample	*	17.95 ± 0.22	12.37 ± 0.3	1.23 ± 0.01	2.39 ± 0.03	balance
Implant	*	17.2 ± 0.3	11.4 ± 0.2	1.75 ± 0.05	2.2 ± 0.15	balance

* Beyond the EDS detection range.

**Table 2 materials-13-04893-t002:** Parameters for the additive manufacturing by LENS method of a sample made of SS316L.

Laser Power (W)	Head Feed Rate (mm/s)	Powder Feed Rate (g/s)	Layer Thickness (mm)	Hatch Width (mm)
375	8	0.12	0.3	0.4

**Table 3 materials-13-04893-t003:** Proof stress values for selected of the 316L alloy made by LENS technology.

Sample No.	Strain Rate (s^−1^)	σ_5%_ (MPa)	σ_15%_ (MPa)
1	2870	820	941
2	2850	797	914
3	2830	794	912
4	2810	819	959
5	2840	821	967
Average	2840	810.2	938.6

**Table 4 materials-13-04893-t004:** Plastic flow stress for 3% strain of the 316L material.

SS316L Alloy			316L LENS		
Test No.	Strain Rate (s^−1^)	σ_3%_ (MPa)	Test No.	Strain Rate (s^−1^)	σ_3%_ (MPa)
1	620	999	5	970	665.5
2	830	1070	6	1180	695
3	1510	1144	7	1700	738
4	3260	1356	8	2830	772
			9	4100	882

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
