# Peer review of "Microstructure Evolution of 316L Steel Prepared with the Use of Additive and Conventional Methods and Subjected to Dynamic Loads: A Comparative Study"

_materials, 2020, doi:10.3390/ma13214893_

Round 1

Reviewer 1 Report

Chap.: 1. Introduction:

-There is missing a new data on the impact of additive technologies on mechanical properties, e.g. publication of authors: Petrousek p., et al: Acta Metallurgica Slovaca, vol. 25, 2019, no. 4, p. 283-290, DOI 10.12776 / ams.v25i4.1366, have shown that it is possible to obtain higher properties than described in this article.

Chap.: 2. Materials and Methods 

i) This chapter has a non-standard character without precise description of experimental methods, e.g. this is missing the detail description of forging conditions.

ii) Tab.1: Chemical composition must be given as a local spectral chemical analysis with precise definition of each elements. Also is missing detail chemical analysis for bulk and PM material AISI 316L.

Line 197: Tab.4: Experimental conditions from point of view strain rate are different for the material as bulk and as PM material AISI 316L.

Line 205: Authors have to prove existence of dynamic recrystallization.

Line 223: The authors statement regarding to mechanism of plastic deformation running by twinning and rotation of crystals must be explaining in details on the base of calculation of stacking fault energy and stress for starting of twinning.

Fig.9e: Dynamic recrystallization cannot be recognized from this figure.

Line 266: Dynamic recrystallization cannot increase the share of LAGB!

Line 300: What identification method was used to identify ferrite in the structure?  The authors didn’t observe α ′martensite in the structure?

Chap.: 4. Discussion. This chapter has the character of a simple and general processing without analysis of mechanical properties based on the development of the structure, processing conditions and evaluation of fundamental differences between bulk and PM material.

Author Response

Dear Reviewer,
In the attached file I am sending responses to the review.

Reviewer 2 Report

please see the attachment (review report)

Author Response

(The authors gave the same response as above.)

Reviewer 3 Report

  1. The division sign is used in many places in the article (exp. Page 2, line 90: “with a spatial resolution of 30÷50 nm”), should it be changed to “…30–50 nm”.
  2. The strain rates of the comparative test materials are not consistent (results listed in Fig. 3 and Table 4). Please explain whether this is caused by the test properties of the instrument itself or caused by the inconsistency of the initial experimental conditions.
  3. Page 8, line 228: “Fig. 4df” should it be changed to “Fig. 4d–f”
  4. How to verify that LENS technology produces grain boundary ferrite? Are there other experiments such as SEM or OM pictures to verify it?
  5. How to confirm that the two comparative materials in Figure 11 have formed twins and slip/shear bands respectively after deformation, the morphology of the two is not much different seen from the images. It is necessary to supplement literature or experimental data to explain it.

Author Response

(The authors gave the same response as above.)

Reviewer 4 Report

Research on additively manufactured metals continue to attract high interest from the academe.

I think the paper may still be improved, please see my comments in the attached pdf.

Some major points to be addressed:

(i) In your introduction, I think you need to put better perspective on the significance of the study. Mention why it is important to do dynamic strain rate testing on an AM specimen in the first place. Others would have the opinion that AM specimens are visual prototypes only, or that it would not be shaped by further wrought processing, so dynamic loading may be useless. Hence, try to point out why doing this test is important and how tracking microstructural  evolution during dynamic loading bears significance.

(ii) What is grossly lacking is the absence of a discussion on the implications of your work. I find the discussion a bit shallow and merely summarizes a lot of the result. I think if you can highlight the significance in the introduction (as suggested above), you can then add meaningful discussions on the implications of your results. You may also cite other works, and explain how different your results are from these.

Author Response

(The authors gave the same response as above.)

Round 2

Reviewer 4 Report

I commend the authors for satisfactorily addressing the concerns on the manuscript.